# Selection of Chemotherapy in Advanced Poorly Differentiated Extra-Pulmonary Neuroendocrine Carcinoma

**DOI:** 10.3390/cancers15204951

**Published:** 2023-10-11

**Authors:** Jamie M. J. Weaver, Richard A. Hubner, Juan W. Valle, Mairead G. McNamara

**Affiliations:** 1The Christie NHS Foundation Trust, University of Manchester, Wilmslow Road, Manchester M20 4BX, UK; jamie.weaver2@nhs.net (J.M.J.W.); richard.hubner@nhs.net (R.A.H.); juan.valle@nhs.net (J.W.V.); 2Division of Cancer Sciences, School of Medical Sciences, University of Manchester, Manchester M20 4BX, UK

**Keywords:** neuroendocrine carcinoma, chemotherapy, poorly differentiated, extra-pulmonary

## Abstract

**Simple Summary:**

Extra-pulmonary poorly differentiated neuroendocrine carcinoma is a rare tumour type with a limited evidence base for its treatment. Recent work has helped to clarify the optimum first-line chemotherapy regimen. However, in the second-line setting, data remain sparse. A more personalised approach is warranted, given the heterogeneity of this disease, and emerging translational approaches focused on mouse models, organoids, and comprehensive genomic profiling may guide future trial design.

**Abstract:**

Extra-pulmonary poorly differentiated neuroendocrine carcinoma is rare, and evidence for treatment has been limited. In this article, the evidence behind the cytotoxic chemotherapy choices used for metastatic or unresectable EP-PD-NEC is reviewed. In the first-line setting, etoposide and platinum chemotherapy or irinotecan and platinum have been demonstrated to be equivalent in a large phase III trial. Questions remain regarding the optimal number of cycles, mode of delivery, and the precise definition of platinum resistance in this setting. In the second-line setting, FOLFIRI has emerged as an option, with randomized phase 2 trials demonstrating modest, but significant, response rates. Beyond this, data are extremely limited, and several regimens have been used. Heterogeneity in biological behaviour is a major barrier to optimal EP-PD-NEC management. Available data support the potential role of the Ki-67 index as a predictive biomarker for chemotherapy response. A more personalised approach to management in future studies will be essential, and comprehensive multi-omic approaches are required to understand tumour somatic genetic changes in relation to their effects on the surrounding microenvironment.

## 1. Introduction

Extra-pulmonary poorly differentiated neuroendocrine carcinoma (EP-PD-NEC) is a rare disease with a poor prognosis. Neuroendocrine neoplasms form a continuum of biology ranging from well-differentiated, slowly proliferating lesions—neuroendocrine tumours (NETs)—to poorly differentiated, faster proliferating large or small cell neuroendocrine carcinomas (NECs) [1]. Poorly differentiated (PD) tumours are characterised by the presence of significant nuclear atypia patches of necrosis and grossly distorted tissue architecture. The majority of EP-PD-NECs arise in the gastrointestinal tract, but they can arise from a variety of other organs, with up to a third having no identifiable primary site [2]. In addition, most cases present at an advanced disease stage where cure is not an option, with a median prognosis of usually less than 1 year [3].

The genetic or epigenetic changes that drive this biology have begun to be determined through detailed genomic sequencing work [4]. Higher-grade poorly differentiated tumours more commonly harbour TP53 mutations and RB1 loss, in keeping with those seen within pulmonary small cell carcinoma [5]. Recurrent mutations in many genes have been identified, but, as of yet, no highly recurrent targetable driver genes have been identified [6,7]. Rare instances of druggable targets are emerging, such as microsatellite instability, Neurotrophic tyrosine receptor kinase (*NTRK*), and v-raf murine sarcoma viral oncogene homolog B1 (*BRAF)*, which are enriched in NECs originating from the colon [4]. Nonetheless, cytotoxic chemotherapy currently remains the mainstay of clinical management in the metastatic setting, except in rare circumstances where tumour-agnostic therapies are indicated, as above.

In high-grade neuroendocrine neoplasms, patient prognosis is closely linked to morphological findings upon pathological assessment, including the degree of differentiation and large-cell or small-cell morphology within the poorly differentiated cohort [8]. Patients with poorly differentiated carcinomas with increased proliferation rates have significantly worse outcomes than those with well-differentiated tumours with similar proliferation rates [9]. The assessment of neuroendocrine neoplasm morphology is complex, and given the prognostic and predictive implications, centralized specialist pathology review is essential (Figure 1). It is important to note that prognosis has also been associated with the proliferation rate, measured either by mitotic index or by Ki-67 staining; cases with higher Ki-67 indices have worse overall survival [10]. A higher Ki-67 index has also been shown to be predictive of the response to therapy. A cut-off of the Ki-67 index of >55% has been suggested to identify EP-PD-NEC cases with a better response to first-line therapy [10]. However, in a recent meta-analysis, this relationship was not present in the second-line setting [11]. Furthermore, a recent, detailed study of pancreatic NEN after expert pathologist assessment to separate G3 NETs from NECs also did not show a Ki67 index > 55% to be predictive of the response to platinum-based therapy [6] Further studies are required to better characterise this association in EP-PD-NEC [12].

Given its rarity and biological complexity, data to guide the use of systemic anti-cancer therapy in EP-PD-NEC were previously limited mostly to retrospective case series or single arm prospective phase 2 trials [11]. Recently, however, randomised clinical trial data have begun to emerge, with the presentation of the TOPIC-NEC, BEVANEC, and NET-02 trials demonstrating the feasibility of clinical trials in this disease group when delivered through large specialist networks [3,13,14]. In this review, the current evidence base for the use of cytotoxic chemotherapy in advanced EP-PD-NEC will be explored, with a focus on a proposed management algorithm (Figure 1). The roles of immunotherapy and targeted agents have been discussed in detail in several recent reviews, and they will not be explored in detail in this current manuscript [4].

**Figure 1 cancers-15-04951-f001:**
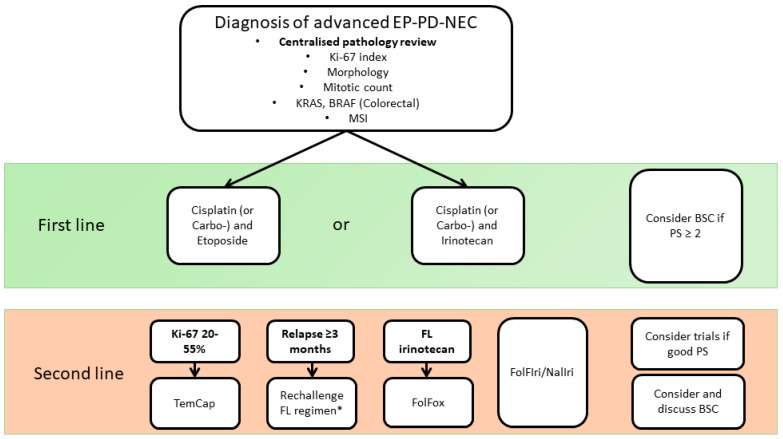
A suggested treatment algorithm. Choice of regimen should also consider patient factors, such as comorbidities and other organ dysfunction. The use of first-line regimen re-challenge with relapse between 3 and 6 months post-first-line should be based on the initial depth of response and the patient’s tolerance of the first-line regimen. TemCap, temozolomide and capecitabine; FOLFIRI, 5-Fluorouracil and irinotecan; NALIRI, nanoliposomal irinotecan and 5-Fluorouracil; BSC, best supportive care; FOLFOX, 5-Flurouracil and oxaliplatin; FL, first line; PS, ECOG performance status [10,15,16,17]. * Depth of response and tolerance to first-line therapy along with comorbidities are important considerations in deciding on re-challenge.

## 2. First-Line Systemic Therapy Options for Patients with Advanced EP-PD-NEC

A choice of first-line therapy in patients with EP-PD-NEC is a platinum, either carboplatin or cisplatin, in combination with etoposide [18]. This was first proposed due to the perceived similarity of EP-PD-NECs and small cell carcinomas of the lung, which have an extensive evidence base. The efficacy of this treatment was first demonstrated in a cohort of 18 tumours, defined by histopathological morphology criteria as EP-PD-NEC-NEC [19]. Responses were seen in a total of 12/18 (67%) patients, with a complete response (CR) in 3 cases. A subsequent study demonstrated a response rate of 41.5% (17/41) with the cisplatin/etoposide combination using identical dosing [20]. More recent larger cohort studies have confirmed the efficacy of this first-line regimen; Sorbye et al. reported a response rate of 28% amongst 148 patients with non-small cell EP-PD-NECs, with a progression-free survival (PFS) of 4 months and OS of 11 months [10]. The same study reported an additional 118 patients with small cell morphology, but no difference was observed in response rate or OS in this population. Yamaguchi et al. reported a response rate of 28% amongst 46 patients treated with EP in a Japanese population in the first-line advanced setting, with an OS of only 7.3 months, and Frizziero et al. reported a disease control rate for platinum/etoposide of 74.5% with a median OS of 11.5 months [21,22].

Guidelines support the use of cisplatin or carboplatin in the first-line advanced setting for EP-PD-NEC in combination with intravenous or oral etoposide [23,24]. In the Nordic-NEC database, no difference was observed in ORR or OS between patients treated with carboplatin and cisplatin [10]. In the only randomised phase 3 trial to date, TOPIC-NEC, cisplatin was used in both arms, and in a recent survey of 75 oncologists treating EP-PD-NEC, there was equipoise, with 52% favouring cisplatin [25]. In small cell lung cancer, a meta-analysis of four trials, including a total of 663 patients (453 extensive stage-SCLC), confirmed that cisplatin and carboplatin provided equivalent benefits, with no difference seen in the Objective Response Rate (ORR) (67% vs. 66%, cisplatin vs. carboplatin) and no difference in OS [17]. Myelotoxicity was increased with carboplatin; in particular, thrombocytopenia, nephrotoxity, ototoxity, emesis, and alopecia were increased with cisplatin. There was no toxicity-associated mortality difference between the two agents. An additional single retrospective study has compared the two platinum agents in 263 patients with EP-PD-NEC in the first-line advanced setting, suggesting carboplatin was associated with worse OS in a Canadian population. This association remained significant in multivariable analysis, while controlling for disease stage (Hazard Ratio (HR) 1.40; 95% Confidence Interval (CI) 1.02–1.92); however, the retrospective nature of the study means that bias in patient selection is likely [26]. The use of cisplatin or carboplatin in the first-line setting should be individualized to the patient, with consideration given to age, renal function, and performance status. In SCLC, it has been demonstrated that there is no benefit in survival with the use of the intravenous (IV) route versus the oral route for etoposide [27]. In EP-NEC, two retrospective studies have addressed this question. In 236 patients from the NORDIC-NEC register focusing on gastroenteropancreatic (GEP) neuroendocrine neoplasms (NENs), no difference was observed between IV etoposide and oral etoposide, although only 33 patients received the oral formulation [28]. Similarly, in 113 patients with GEP-NEC, Frizziero et al. showed no difference between OS and ORR between oral and IV administration, despite a large numerical difference in OS (8.9 vs. 12.1 months, respectively) [21]. Given documented patient preferences for oral regimens, oral etoposide in some cases may be the more optimal route of delivery [29].

Alternative regimens to carboplatin/etoposide in the first-line advanced setting are advocated by some guidelines. A recent phase 3 trial run across 50 centres in Japan compared cisplatin (80 mg/m^2^) and etoposide (100 mg/m^2^ D1, 2 and 3) given in 3-weekly cycles with irinotecan (60 mg/m^2^ D1, 8 and 15) and cisplatin (60 mg/m^2^, D1) given every four weeks [3]. In total, 170 patients with digestive NEC were randomised, 50 percent of which were small cell carcinomas and nearly 40% of which were oesophageal or gastric in origin. Central pathology review was mandated and performed on all patient tumours, where blocks were available (168/170); 9.5% of patients were reclassified. Treatment was continued until progression or intolerable toxicity, with a median of 4.5 cycles given in each arm. Response rates were the same with the two regimens (etoposide/cisplatin vs. irinotecan/cisplatin, 54.5% vs. 52.5%, respectively). Median OS was not significantly different between arms (etoposide/cisplatin vs. irinotecan/cisplatin, 12.3 months vs. 10.5 months, respectively). There was no difference in survival between sub-groups, except for a possible benefit for etoposide/cisplatin within patients with pancreatic PD-NECs, though numbers in this group were small (*n* = 23). There was an excess of bone marrow toxicity in the etoposide/cisplatin group, with significantly higher rates of febrile neutropenia. However, with the introduction of primary prophylaxis with granulocyte-colony stimulating factor (G-CSF), this was mitigated. The only treatment-related death occurred in the irinotecan/cisplatin arm. This trial therefore provides support for the current standard choice of etoposide/cisplatin in the first-line advanced setting, with irinotecan/cisplatin being an acceptable alternative.

Consensus on the number of cycles is limited across guidelines. Data from SCLC support the use of four cycles only with a single small phase 3 trial, with 46 ES-SCLC patients showing no benefit to extending treatment to six cycles vs. four cycles [30]. This has been confirmed by a large single-site United-Kingdom-based retrospective study that found no difference in OS between those patients treated with four cycles and those treated with six [31]. Therefore, four cycles of etoposide/cisplatin may be an option for EP-PD-NEC also. Nonetheless, given data specific to EP-PD-NEC are not available, there remains a need for an expert consensus view on this topic [25].

Though response rates to first-line platinum doublets are high, PFS is short, as discussed above, and alternative regimens are being explored. A recent trial (NCT02595424) has investigated the efficacy of temozolomide and capecitabine versus carboplatin etoposide (EP) as a first-line treatment [32]. Both regimens provided similar outcomes, with low response rates of 9% and 10%, respectively. The trial included both G3 NEC and NETs, with only 57 percent of tumours being poorly differentiated, and a relatively low overall Ki67 index of 48% in the TemCap arm and 60% in the EP arm. Further details on response rates by sub-group will be important to better understand the clinical relevance of these findings.

Given the limited response rates to first-line doublet therapy, triplet therapies have been explored. In a small retrospective study, a triplet regimen of 5-flourouracil, irinotecan, and oxaliplatin has shown an ORR of 46% [33]. Only eight patients in this study were treated in the first-line setting, and a detailed breakdown of the sub-group ORR was not available. The prospective FOLFIRINEC trial comparing Folfirionx to first-line EP chemotherapy will clarify the role of this regimen [34]. A triplet regimen of paclitaxel, etoposide, and platinum therapy demonstrated an ORR of 53% in a single arm study comparable to doublet therapy alone; therefore, this regimen has not been pursued further [35].

## 3. Second-Line Treatment for EP-PD-NEC in the Advanced Setting

Until recently, consensus for second-line treatment was unclear in this setting. The recent presentation of the prospective BEVANCEC (Bevaciuzumab added to FOLFIRI) and NET-02 (liposomal irinotecan with 5-flurouracil versus Docetaxel) randomised phase 2 trials has helped to somewhat clarify the options; however, definitive phase 3 data are awaited [13,14]. Overall, a relatively small proportion of patients with advanced EP-PD-NEC receive second-line treatment [21]. This is likely due in part to the aggressive nature of the disease and the limited evidence of efficacy of second-line regimens. More robust prospective data are therefore needed.

### 3.1. Re-Challenge in the Second-Line Setting

When deciding on treatment choice in this setting in patients who are fit for second-line therapy, it is important first to determine whether resistance to platinum therapy has developed. As with SCLC, time cut-offs are implemented to make this decision. A recent consensus from 75 NET specialists found the favoured re-challenge would be with carboplatin/etoposide if progression occurred after 6 months [25]. No study has looked prospectively at this question, but in two retrospective case series, response rates to second-line platinum re-challenge were 17% and 31%, though with small numbers in each study [21,22]. Evidence for irinotecan/platinum in the second-line re-challenge setting is extremely limited, with five cases reported in Yamaguchi et al. with a response rate of 40% (*n* = 2), suggesting it may be a suitable alternative to etoposide/platinum while acknowledging the small numbers of patients included [22].

In patients with SCLC of the lung, phase 3 randomised controlled trial (RCT) evidence supports re-challenge if there is progression beyond 3 months. Carboplatin and etoposide re-challenge was compared to topotecan monotherapy in patients who had relapsed beyond 90 days [36]. In a multivariable analysis of factors potentially associated with PFS, treatment-free interval (TFI) (90–180 days versus > 180 days) was not a significant factor. Hadoux et al. conducted a retrospective multicentre study of EP-PD-NEC identifying 94 patients with relapse beyond 90 days [37]. Patients who received a re-challenge had an improved OS compared to those who did not, with no reduction of benefit in those re-challenged between 90 and 180 days when compared to those re-challenged beyond 180 days (12 vs. 5.9 months, *p* = 0.043). However, these data did include thoracic NEC cases; therefore, the implications for EP-PD-NEC remain unclear. Larger case series, or, preferably, prospective studies, of patients with EP-PD-NECs will be required to determine if patients with a shorter TFI may also benefit from re-challenge and, in particular, if this applies across the proliferation and morphological spectrum of this disease.

For patients with EP-PD-NEC relapsing with platinum resistance, the evidence base for second-line options is limited to retrospective case series and, more recently, phase 2 trials. A meta-analysis of the available literature in the second-line advanced setting identified a total of 15 different regimens used across 19 studies of EP-PD-NEC [11]. Given the poor prognosis of patients with platinum-resistant EP-PD-NEC, it is essential that the benefits of any regimen are always weighed against the toxicities, and consideration should always be given to the best supportive care as an appropriate approach, particularly in those patients with poor PS, to maximise patient quality of life.

### 3.2. Single-Agent Chemotherapy in the Second-Line Setting

The regimens used in the second-line clinical setting cover a variety of different cytotoxic chemotherapy agents, with different targets that have been used as monotherapy or in doublets. The topoisomerase I inhibitors, topotecan and irinotecan, have both been assessed as monotherapies in several retrospective studies, including in patients with an EP-PD-NEC diagnosis. However, overall efficacy has been limited. In one retrospective assessment of topotecan in 30 patients, 2 (7%) had a partial response, whilst in a study of 22 patients, the ORR was 0% [38,39]. This response rate is similar to that seen with irinotecan monotherapy in this setting, with only 5 of 21 patients having a partial response in a recent retrospective analysis [22]. Though low, these rates are broadly comparable to the response rate seen in the registration phase 3 trial in SCLC of single-agent topotecan in the second-line advanced setting, where ORR was 7% (5 of 71). Despite this poor ORR, a near doubling of OS was observed [40]. Nonetheless, monotherapy with topoisomerase I inhibitors is not recommended in international guidelines for the treatment of patients with EP-PD-NEC [18]. Amrubicin, a topoisomerase II inhibitor, has also been trialled in several small case series (12–16 patients) in EP-PD-NEC at doses between 30 and 45 mg/m^2^. Response rates have been better than that seen with single-agent topotecan, with RRs ranging between 6.3% and 50% and PFS between 2 and 6 months [15,22,41,42]. This increased RR was associated with significant myelotoxicity, with Grade 3–4 neutropenia rates of 50–60% across all three studies. To the author’s knowledge, only one prospective phase II study of single-agent topoisomerase inhibitors in EP-PD-NEC has been published. In this study, 22 patients, over half of whom had a Ki-67 > 55%, were treated with a novel topoisomerase I inhibitor TLC388 at 40 mg/m^2^ [43]. Disappointingly, no responses were seen, and PFS was only 1.8 months, despite the improved in vitro data showing higher rates of topoisomerase I inhibition when compared to topotecan [44]. Single-agent topisomerase inhibitors do not, therefore, have a role currently in the management of patients with EP-PD-NEC.

Alkylating agents have also been assessed as monotherapy in the second-line setting in small prospective and retrospective studies that included patients with EP-PD-NEC. Kobayashi et al. conducted a randomised phase II trial in patients with EP-PD-NEC with temozolomide (200 mg/m^2^) given on days 1–5 of 28-day cycles. Responses were seen in 2 of 13 patients. Interestingly, both patients had higher proliferation rates (defined in the study as Ki-67 >50%) [45]. This confirms the findings of an earlier retrospective study, which also showed low response rates with monotherapy with temozolomide of 0% and a PFS of only 2.4 months in 16 patients with evaluable disease. Again, in this study, more than half of patients had a Ki-67 index ≥ 50%. The results of the second-line TENEC trial in the advanced setting are awaited to clarify the benefit of single-agent temozolomide in patients with EP-PD-NEC, and, in particular, the predictive value of the Ki-67 index (NCT04122911).

Given its efficacy in small cell lung cancer, Lurbinectedin, an inhibitor of oncogenic transcription, has been investigated as an option in NEN. A recent basket trial of second-line therapy including a heterogenous population of NENs, including NETs and NECs, showed PR in two out of thirty-two patients, one of which was a high-grade NEC. Without further information on subtypes, it is difficult to draw further conclusions about the efficacy of this therapy option in this setting [46].

The use of single-agent antimetabolite drugs in this setting has been explored less. S-1 is an antimetabolite cancer agent composed of tegafur 5-chloro-2, 4-dihydroxypyridine, and oteracil potassium. In 11 patients in the second-line advanced setting, it demonstrated an RR of 27%, with a PFS of only 2.8 months [22]. Given the relatively poor RR seen overall with monotherapies in EP-PD-NEC, doublet therapies have also been explored in the second-line setting.

### 3.3. Combination Chemotherapy Regimen in the Second-Line Setting

Given the comparable efficacy of first-line platinum/etoposide and platinum/irinotecan that can be seen in TOPIC-NEC, the use of irinotecan in doublets in the second-line setting in advanced EP-PD-NEC has been explored in several studies, and liposomal irinotecan has been examined in one prospective study. In a small retrospective study including 16 patients treated with irinotecan (180 mg/m^2^) and 5-fluorouracil (2000 mg/m^2^) in the second-line advanced setting, the RR was 33%, with a PFS of 4 months [47]. The BEVANEC phase II trial investigated the use of bevacizumab with chemotherapy in EP-PD-NEC and utilized FOLFIRI (Irinotecan 180 mg/m^2^, 5-FU, 2800 mg/m^2^) as its control arm with an ORR of 18% (11 of 61 patients). The majority of patients in this study had a Ki-67 index ≥ 55% (81%). Another prospective phase II study, NET-02, recruited 102 patients and randomised them 1:1 to liposomal–irinotecan (70 mg/m^2^) and 5-flourouracil (2400 mg/m^2^) IV every two weeks, or docetaxel IV every three weeks (75 mg/m^2^), until progressive disease or intolerance. Patients in this study were exclusively poorly differentiated and had grade 3 disease, with 90% having a Ki-67 index ≥ 55% and 91% being platinum resistant. In this clearly defined population, the ORR was 11.1% with liposomal irinotecan and 5-FU and 10.3% with single-agent docetaxel. Despite comparable RRs, the primary endpoint, the 6-month PFS rate, was nearly doubled in the combination arm (29.6% vs. 13.8% for liposomal irinotecan and 5FU vs. docetaxel). Importantly, grade 3 or 4 events were numerically less common in the doublet arm (51.7% vs. 55.2%). However, there was no difference in final OS between the arms, though the study was not powered to detect this [13].

The use of oxaliplatin in combination with 5-fluorouracil (5-FU) has also been explored in patients with advanced EP-PD-NEC. Hadoux et al. treated 20 patients with a two-weekly regimen of oxaliplatin (85 mg/m^2^) and 5-FU (2800 mg/m^2^) in the second-line advanced setting. Seventeen patients were evaluable, and responses were seen in 29%, including those who had not previously responded to first-line platinum therapy, with a median PFS of 4.6 months [48]. Further prospective trials may be warranted.

Given its efficacy in lower-grade NETs, particularly those of pancreatic origin and the relatively poor RR to single-agent temozolomide in EP-PD-NEC, temozolomide and capecitabine have also been investigated as a regimen. Thomas et al. performed a retrospective review of a diverse range of NETs treated with capecitabine and temozolomide. Of 26 patients treated in varying lines who had Ki-67 indices ≥ 20%, partial responses were seen in 12%. In the multivariable analysis, poor differentiation across all NENs was an adverse factor for response, but precise details on differentiation rates and RR were not given in this study [49]. In a study of patients with grade 3 NENs, including well-differentiated and poorly differentiated cases, an RR of 26% was seen amongst 46 patients with EP-PD-NECs in the second-line advanced setting [50]. Response rates in EP-PD-NEC were lower than those seen in well-differentiated cases (26% vs. 41%, respectively), and they were higher in EP-PD-NECs with Ki-67 < 55% vs. those ≥ 55%, in contrast to the pattern seen with first-line chemotherapy with etoposide and platinum [51]. The ongoing second-line phase II trial, SENECA, will prospectively compare temozolomide and capecitabine to 5-FU/irinotecan in second-line patients with EP-PD-NEC, and the comparison will be informative for future trial design. MGMT methylation has been suggested as a biomarker of response to temozolomide in neuroendocrine neoplasms. However, study results have varied, and limited evidence is available for NECs in particular; as such, currently, it cannot be recommended as a standard of care biomarker [45,52,53]. Pancreatic origin has also been shown to be predictive of better response to alkylating chemotherapy in lower-grade and well-differentiated NENs, and differences in MGMT promoter methylation may underly this. Nonetheless, the role of the organ of origin is less clear in poorly differentiated carcinomas [52,53].

In summary, the evidence base for second-line chemotherapy in patients with advanced EP-PD-NEC is limited (Table 1). Guidelines recommend regimens that have comparable efficacy in retrospective cohort studies and small phase 2 trials [18]. Given its limited toxicity and efficacy in a randomised phase 2 trial, 5-FU/irinotecan is an option for second-line treatment in patients with grade 3 EP-PD-NEC, particularly in those with Ki-67 indices above 55%. An option beyond that may be 5-FU/oxaliplatin, best supportive care, or clinical trials, if available.

## 4. Discussion

### Improving on and Moving beyond Current Chemotherapy Regimens in Patients with EP-PD-NEC

Given the poor prognosis for patients with advanced EP-PD-NEC, it is clear that the treatment paradigm for these patients needs to improve. Though responses in the first-line advanced setting are common, rapid progression is the norm, and OS remains less than a year from the start of treatment [10]. The heterogeneity of this disease is striking, and recent developments and clarifications to classification confound the interpretation of many earlier trials and retrospective reviews. The predictive power of the Ki-67 index and morphology have become increasingly clear, and they are essential factors to be incorporated into inclusion criteria in future trial design [54]. Given the available evidence for differentiation and Ki-67 as predictive biomarkers of cytotoxic chemotherapy efficacy in patients with grade 3 NENs, it is essential that future trial outcomes are reported clearly for these relevant sub-groups.

In summary, however, the rate and duration of response to current cytotoxic regimens is poor, particularly in the second-line setting [11]. Furthermore, end-organ dysfunction can limit treatment options in relapsed disease, and consideration must be given to the safety and toxicity of regimen choices. In particular, liver dysfunction can limit the suitability of second-line irinotecan, and peripheral neuropathy may limit the use of an oxaliplatin-based regimen. Further studies of alternative systemic anti-cancer therapies alone or in combination with cytotoxic chemotherapy are warranted. Initial trials of angiogenesis inhibitors have disappointed, despite strong pre-clinical evidence. The BEVANEC phase 2 trial in EP-PD-NEC, predominantly with Ki-67 > 55% (85%), demonstrated that the addition of bevacizumab (5 mg/kg) to FOLFIRI did not improve the response rate (25.5% vs. 18.3%) or OS at 6 months (53% vs. 60%) [14].

The option for immunotherapy in patients with EP-PD-NEC has received much interest more recently. In small cell cancer of the lung, two phase 3 trials have shown an improvement in OS with the addition of Programmed cell death protein 1 (PD-1)/Programmed Cell Death Ligand 1 (PD-L1) directed therapy in combination with first-line chemotherapy [55,56]. A phase 3 trial is currently investigating the addition of atezolizumab to chemotherapy in the first-line advanced setting in patients with EP-PD-NEC (Table 2). Recent small phase II studies in patients with EP-PD-NEC suggest relatively limited RR to single-agent immunotherapy in the second-line setting [57,58,59]. Treatment with dual check point blockade (PD-1 and Cytotoxic T-lymphocyte-associated protein 4 (CTLA-4)) has demonstrated greater efficacy, with an ORR of 26–44% in patients with previously treated high-grade NENs [60,61]. Larger studies are awaited to validate this finding.

Clearly, novel therapies are required to improve outcomes in this disease, and a better understanding of its biology and, in particular, the somatic genetic drivers is urgently needed. Van Riet et al. completed one of the most comprehensive studies to date of high-grade extra-pulmonary neuroendocrine tumours, performing whole genome sequencing (WGS) of 85 cases, including NET and NEC (*n* = 15) samples [5]. They confirmed the striking difference in genomic profiles between G3 NETs and G3 NECs, with elevated point mutation rates and increased aneuploidy and chromosomal rearrangements in NEC cases. However, despite the comprehensive nature of WGS, only a fraction of cases harboured currently targetable genomic alterations, including small numbers of cases with tumour mutation burden (TMB) > 10 per MB and cases with KRAS and PIK3CA mutations. In NEC of colorectal origin, there appears to be a greater proportion of cases with targetable drivers, with a high prevalence of BRAFV600E mutations and microsatellite instability [16,62,63,64,65]. BRAF inhibitors in combination with MEK inhibitors have shown promise in isolated case reports of patients with EP-PD-NEC and BRAFV600E mutations [62,65]. Interestingly, unlike in colorectal adenocarcinomas, significant responses are seen without the need for combination with EGFR inhibition [63]. Whether BRAF inhibitors and/or immunotherapy will be more broadly successful in patients with EP-PD-NEC, as seen in colorectal cancer, awaits further large prospective clinical trials. Larger multi-omics studies incorporating NECs from a wide range of organs will be essential to identify new potential therapeutic avenues. Recent ENETs guidelines suggest comprehensive molecular profiling for NECs where available. Given the limited tumour agnostic approvals for many medications and the tissue specificity of actionable variants, e.g., BRAFV600E in colorectal NEC, the benefit of testing all NECs remains unproven. A limited gene-panel testing approach in tumours arising from specific sites is likely to provide similar benefits. A lack of suitable models for translational work has also hampered progress in the EP-PD-NEC field. Recently, it has been demonstrated that organoids can be established from EP-PD-NECs that recapitulate the genomic and transcriptomic findings seen in primary tissue [66]. The exploration of alternative strategies for model generation (for example, circulating tumour-cell-derived xenografts) is also required to ensure models are generated from chemotherapy-resistant cases, among other settings, as well as pre-treatment cases [4].

Given the rarity of EP-PD-NEC, large national and multinational collaborations and prospective trials are central to the understanding of its biology and in providing valuable insight into future therapeutic options. Given the paucity of identified targetable drivers, cytotoxic chemotherapy will likely continue to play a key role in future trials. Understanding the optimal regimens for each line of therapy is therefore crucial to future management and research into this tumour type.

## 5. Conclusions

The prospective data for the choice of chemotherapy regimen in patients with advanced EP-PD-NEC still lag behind those seen in many other cancer types. In large part, this has been due to the comparative rarity of this tumour and the heterogeneity of its biology. A better understanding of the biology of this disease has made it clear that simple biomarkers, including morphology (small cell versus large cell) and proliferation markers (Ki-67 <55% vs. >55%) have significant predictive power for determining the efficacy of chemotherapy. Future trials using appropriate stratification and large multi-national collaborations may help to clarify the most suitable chemotherapy regimens. Crucially, these trials may provide the ideal background for translational studies to identify the next generation of precision medicine approaches to potentially improve outcomes for patients with EP-PD-NEC.

## Figures and Tables

**Table 1 cancers-15-04951-t001:** Some selected ongoing chemotherapy trials in patients with EP-PD-NEC.

Clinicaltrials.gov Trial Identifier	Trial Regimen	Line of Treatment	Phase	Recruitment Target	Primary Endpoint
NCT02595424	TEMCAP vs. EP	First	II	59	PFS
NCT05058651	Atezolizumab + EP vs. EP	First	II/III	189	OS
NCT04325425	mFolFIrinOx vs. EP	First	II	218	PFS
NCT04042714	TAS-102	Second	II	14	ORR
NCT03387592	FolFIri vs. TEMCAP	Second	II	112	DCR/AE incidence

TEMCAP, temozolomide and capecitabine; FolFIri, 5-flurouracil and irinotecan; TAS-102, trifluridine and tipiracil hydrochloride; mFolFIrinOx, modified 5-Flurouracil, irinotecan, and oxalplatin; EP, cisplatin or carboplatin and etoposide; DCR, disease control rate; OS, overall survival; PFS, progression-free survival; AE, adverse event.

**Table 2 cancers-15-04951-t002:** Published randomised control trials in first- and second-line treatment of EP-PD-NEC.

Trial ID	Trial Regimen	Line of Treatment	Phase	Recruitment Numbers	Main Primary Sites	Ki-67 ≥ 55%	ORR	PFS (Months)	OS
TOPIC-NEC	EP vs. IP	1	III	170	Upper GI	82% vs. 81%	54% vs. 52%	5.6 vs. 5.1	12.5 vs. 10.9 months
Bevanec	FOLFIRI + Bev vs. FOLFIRI	2	II	150	Colorectal	86% vs. 76%	25% vs. 18%	3.7 vs. 3.5	6 months, 53% vs. 61%
NET-02	Nal-IRI/5FU vs. Docetaxel	2	II	59	Upper GI	90% vs. 90%	11% vs. 10%	3 vs. 2	6 months, 29.6% vs. 13.8%

EP, cisplatin or carboplatin and etoposide; IP, cisplatin or carboplatin and irinotecan; FOLFIRI, 5-flurouracil and irinotecan; Bev, Bevacizumab; Nal-IRI/5FU, Nano-liposomal irinotecan and 5-flurouracil; GI, gastrointestinal; ORR, overall response rate; PFS, progression-free survival; OS, overall survival. Note: Ki-67 values for TOPIC-NEC were reported as those patients above 50%, not 55%. OS for BEVANEC and NET-O2 is given as 6-month OS percentage. No significant difference was seen for the comparison of OS, ORR, and PFS in TOPIC-NEC; Bevanec and NET-02 were non-comparative phase 2 studies, so no comparisons were made directly between arms for ORR, PFS and OS.

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
