# Peer review of "Selection of Chemotherapy in Advanced Poorly Differentiated Extra-Pulmonary Neuroendocrine Carcinoma"

_cancers, 2023, doi:10.3390/cancers15204951_

Round 1
Reviewer 1 Report
The present paper is a well-written review describing the role of palliative chemotherapy for patients with poorly differentiated metastatic neuroendocrine neoplasms. This review is very detailed and highlights important issues about the topic. I appreciated that the authors outlined that also best supportive care is an option for patients with poor performance status, especially when considering second-line therapies.
I have only a few minor remarks for the authors:
1) References are not in the correct order (from the first citation to the last, as usual);
2) Page 6, line 251; when discussing NET02 trial results, the authors stated that the absence of difference in overall survival between arms may be due to the lack of effective therapies post second-line. Usually, the opposite is true: effective post-progression treatments dilute the benefits seen in intermediate endpoints, such as progression-free survival in trials evaluating novel therapies in solid tumors. Therefore, the authors should reconsider their claim;
3) Page 8, line 340; suitable models instead of suitable modes (typing error).
Author Response
Many thanks for your review of our paper.
We have made the amendments as you suggested, we agree it is unclear what effect second line therapy efficacy would have regarding OS and have amended this section. Thank you
Reviewer 2 Report
1. General comments--
a. Clarify if the review is on GEPNEC or all ep-NEC (no mention of prostate NEC or cervical NEC, etc.)-- and modify text accordingly (the majority of EP-NEC arise in gastrointestinal tract--but about 1/3 unknown primary and 1/3 other/mostly GU--but no mention of data related to these other sites)
b.Given the focus of the review -- it could be helpful to briefly discuss how end organ disfunction, neuropathy, poor PS, etc. influence choice of therapy in 2L and beyond (given the lack of an obvious winner)
c.Consider commenting on use of DPD, UGT1A1 and/or MGMT to guide choice of therapy (even if the recommendation is not to test-- noting if any recent trials have incorporated such biomarkers). In table listing key trials- -would highlight information related to Ki67, if known
d. clarify if primary tumor site or molecular features influence choice of therapy
e. somewhere in manuscript would discuss the challenges of differentiating G3NET from large cell NEC, the potential role of other regimens besides EP if low Ki67, and the role of path review, etc.
2. Fig 1 and text thoughout-—
a.clarify why MSI testing is just recommended for colorectal NEC (many clinicians test all stage IV EP-NECs given tumor-agnostic approvals), as well as the role of molecular profiling (NGS) for actionable mutations and assessment of TMB; consider emphasizing the role of clinical trials (rather than pooling with BSC).
b.A case is made for rechallenge with IL therapy if TFI >3 mo or >6 mo in the text ; it would be helpful to confirm that fig 1 represents the authors' recommendations (i.e. rechallenge would be considered in a patient with TFI<6 mo)
3. 1L section
a. Line 127 or so—suggest mentioning that >40% of patients had gastroesophageal primary tumors in the TOPIC-NEC trial (approx double what is seen in some countries, e.g. USA)
b. suggest addressing role of 1L immunotherapy/chemotherapy since standard in 1L SCLC and under study in EP-NEC (instead of just mentioning in discussion)
c. consider mentioning EA 2142 (listed in table 1 NCT02595424) as results now available, at least in abstract form
4. 2L section
a. It would help to have separate sections for platinum-sensitive and platinum-resistant disease. Also, consider including subheadings for the different classes of agents, e.g. temozolomide-based therapy (TMZ, TMZ/cape), topoisomerase inhibitors—including topotecan and irinotecan
b. would include information about the efficacy of additional agents/regimens, e.g other taxanes, lurbinectedin (approved in SCLC), and FOLFIRINOX
c. provide reference for the statement, "Overall, a relatively small proportion of patients with advanced EP-PD-NEC receive second-line treatment " (line 155)
5. Discussion- Suggest addressing whether or not the primary site matters, either in terms of biomarker assessment (TMB, MSI, mutation profiling, Ki67, etc) or choice of therapy. Also consider addressing role of molecular profiling and use of agents to treat actionable mutations (and where these might fit in relative to chemotherapy)
6. Conclusion - The conclusion in the text doesn't quite match the focus on FOLFIRI in the abstract (i.e. not clear that we have sufficient data to say that FOLFIRI is the preferred 2L regimen)
7. Additional suggestions
a. suggest adding tables summarizing 1) key 1L trials (e.g. phase, N, primary sites, biomarker info/Ki67, RR, mPFS, OS, etc); 2) key 2L and beyond trials (phase, RR, mPFS, OS, etc);
b. suggest expanding on trials listed in current table 1, including chemo-combination trials (to provide a more complete picture of where the field is going)
Author Response
1. General comments--
a. Clarify if the review is on GEPNEC or all ep-NEC (no mention of prostate NEC or cervical NEC, etc.)-- and modify text accordingly (the majority of EP-NEC arise in gastrointestinal tract--but about 1/3 unknown primary and 1/3 other/mostly GU--but no mention of data related to these other sites):
AUTHORS: Thank you for spotting this lack of clarity. The work relates to all EP-PD-NEC, we have clarified this in the text now with reference to the Dasari et al Seer database paper.
b. Given the focus of the review -- it could be helpful to briefly discuss how end organ disfunction, neuropathy, poor PS, etc. influence choice of therapy in 2L and beyond (given the lack of an obvious winner)
AUTHORS: Thank you, we have added text to the discussion and the figure 1 legend to reflect this important point.
c. Consider commenting on use of DPD, UGT1A1 and/or MGMT to guide choice of therapy (even if the recommendation is not to test-- noting if any recent trials have incorporated such biomarkers). In table listing key trials- -would highlight information related to Ki67, if known
Authors: This is an important point a reference to this has been included in the alkylating section as suggested. Though signifcant data on MGMT methylation is mostly restricted to well differentiated and lower grade tumours we have now made references to this literature and the limited data available for NEC.
d. clarify if primary tumour site or molecular features influence choice of therapy
Authors: We have included information now focussing on the limited information related to this in regard to pancreatic origin tumours responding to alkylating agents. We already have included a section on molecular analysis that details the tissue specificity of relevant molecular changes and the importance of tissue of origin for these tests. within the discussion. As per another of the reviewers points further on we have added text to this section as well and we hope this has improved its clarity.
e. somewhere in manuscript would discuss the challenges of differentiating G3NET from large cell NEC, the potential role of other regimens besides EP if low Ki67, and the role of path review, etc.
AUTHORS: We have amended the introduction to discuss and reference the complexities of pathology review including the references to differentiation and morphology as well as including path review as part of the suggested treatment algorithm and
2. Fig 1 and text thoughout-—
a. Clarify why MSI testing is just recommended for colorectal NEC (many clinicians test all stage IV EP-NECs given tumor-agnostic approvals), as well as the role of molecular profiling (NGS) for actionable mutations and assessment of TMB; consider emphasizing the role of clinical trials (rather than pooling with BSC).
AUTHORS: We have modified the text to reflect this as you point out and we have modified the figure. We recognise that the recent ENETs guidelines suggest a broader testing of MSI in NEC as a possible option but with low evidence. The rates of MSI in other tumour sites is vanishingly low and appears to mirror the rates seen in adenocarcinomas arising from these tissues. We have also modified the figure to make the distinction clear between BSC and trial clearer as well as suggested by the reviewer
b.A case is made for rechallenge with IL therapy if TFI >3 mo or >6 mo in the text ; it would be helpful to confirm that fig 1 represents the authors' recommendations (i.e. rechallenge would be considered in a patient with TFI<6 mo)
AUTHORS: We have added text to figure one and made it clear that a rechallenge can be considered in patients with a TFI of 3 months based on the Hadoux et al data discussed in the text. As per the reviewers other important poitn we have clarified that rechallenge choice is dependant on depth of roginal response, tolerance of therapy and comorbidities.
3. 1L section
a. Line 127 or so—suggest mentioning that >40% of patients had gastroesophageal primary tumors in the TOPIC-NEC trial (approx double what is seen in some countries, e.g. USA)
Authors: We have amended the text to include a reference to this important observation.
b. suggest addressing role of 1L immunotherapy/chemotherapy since standard in 1L SCLC and under study in EP-NEC (instead of just mentioning in discussion)
Authors: We feel this review is best focussed on the chemotherapy backbone options particularly in light of the difficulties drawing direct comparisons with EP-NEC and small cell. We believe the details we have included in the discussion regarding immunotherapy cover the key evidence already and are best kept in the discussion to maintain the focus on the chemotherapy back bone. We would be happy to move those paragraphs earlier if the reviewer thinks it will improve the readability of the manuscript.
c. consider mentioning EA 2142 (listed in table 1 NCT02595424) as results now available, at least in abstract form.
Authors: We have included a discussion of this trials abstract in the first line section of the review now.
4. 2L section
a. It would help to have separate sections for platinum-sensitive and platinum-resistant disease. Also, consider including subheadings for the different classes of agents, e.g. temozolomide-based therapy (TMZ, TMZ/cape), topoisomerase inhibitors—including topotecan and irinotecan
Authors: We have added sub headings to each section of the SL part of the review to enhance readability as suggested here.
b. would include information about the efficacy of additional agents/regimens, e.g other taxanes, lurbinectedin (approved in SCLC), and FOLFIRINOX
Authors: As suggested we have added information in regarding folfirinox discussing the Butt et al abstract and Folfirinec trial. We have also now referenced the limited data on lurbinectedin . The details of the NET-02 trial we hope cover the evidence behind taxanes.
c. provide reference for the statement, "Overall, a relatively small proportion of patients with advanced EP-PD-NEC receive second-line treatment " (line 155)
Authors: A reference to this data has been included now thank you for identifying this issue.
5. Discussion- Suggest addressing whether or not the primary site matters, either in terms of biomarker assessment (TMB, MSI, mutation profiling, Ki67, etc) or choice of therapy. Also consider addressing role of molecular profiling and use of agents to treat actionable mutations (and where these might fit in relative to chemotherapy).
Authors: We have added additional text to the biomarker section of our discussion to clarify the role of primary site. We have also added to our discussion on colorectal origin and BRAFV600E by discussing G12c mutations and MSI which represent the only other significant findings in the large scale sequencing studies that have been reported.
6. Conclusion - The conclusion in the text doesn't quite match the focus on FOLFIRI in the abstract (i.e. not clear that we have sufficient data to say that FOLFIRI is the preferred 2L regimen)
Authors: Despite having the strongest evidence for any regimen in the second line given the phase 2 nature of the trials supporting FOLFIRI we agree with the authors comments and have modified the abstract text to reflect the degree of uncertainty.
7. Additional suggestions
a. suggest adding tables summarizing 1) key 1L trials (e.g. phase, N, primary sites, biomarker info/Ki67, RR, mPFS, OS, etc); 2) key 2L and beyond trials (phase, RR, mPFS, OS, etc);
Authors: We have added in a table including key published randomised trials in the first and second line setting. We have restricted this to published data to allow inclusion of details on subtyping often not present in abstracts.
b. suggest expanding on trials listed in current table 1, including chemo-combination trials (to provide a more complete picture of where the field is going)
Authors: As discussed above we believe this review is best focussed on chemohtherapy backbones. We have provided references to more generalised recent reviews covering combination therapies in more detail.
Reviewer 3 Report
Summary
Weaver et al. summarized a review on the chemotherapy for Extra-pulmonary Neuroendocrine Carcinoma (EP-PD-NEC). While this field has limited evidence, high-quality studies, including phase III trials, have been reported in recent years, highlighting the relatively high importance of this review. However, there are several concerns on this paper with the evaluation of the level of evidence, and major revisions are necessary.
Major points
1. Introduction
Page 2 Line 45-48; The authors have described the significance of TP53 mutations and RB1 loss in poorly differentiated neuroendocrine tumors. While they have cited the paper by van Riet et al., there are other important studies worth considering as well.
One example is the following:
Hijioka et al. Pancreatology. 2020 Oct;20(7):1421-1427. doi: 10.1016/j.pan.2020.07.400. Epub 2020 Aug 8.
2. Introduction
Page 2 Line 62-63; In this section, the authors mention that a Ki-67 index of higher than 55% indicates a good response to first-line chemotherapy for EP-PD-NEC. However, it is important to note that the cited paper by Sorbye et al. is predating the introduction of the NET G3 concept and the current definition of EP-PD-NEC. Furthermore, it should be clarified that the favorable response applies specifically to cytotoxic chemotherapy and does not always imply a good response to all types of chemotherapy (Hijioka et al. Clin Cancer Res
. 2017 Aug 15;23(16):4625-4632. doi: 10.1158/1078-0432.CCR-16-3135. Epub 2017 Apr 28.)
3. Introduction
Page 2 Line 71-75; Regarding the proposed algorithm for chemotherapy in EP-PD-NEC in Figure 1, I cannot agree with its content. Firstly, regarding first-line chemotherapy, there is limited high-quality evidence for Carboplatin + Etoposide/Irinotecan. Most studies are retrospective, and there is a lack of randomized prospective double-blinded trials. the Cisplatin + Etoposide/Irinotecan, which was demonstrated in the TOPIC-NEC study that there was equivalence efficacy. Carboplatin is considered an alternative treatment option in limited situations, such as when Cisplatin is not suitable due to impaired renal function, etc.
For second-line chemotherapy, the algorithm suggests TemCap is effective for Ki-67 indices ranging from 20-54%, however, it should actually be based on a cut-off of 55% and it should be 20-55%. Additionally, I believe that there is limited evidence supporting the efficacy of TemCap specifically for NEC, rather than NET-G3.
4. First line systemic therapy options for patients with advanced EP-PD-NEC
Page 3 Line 83; As mentioned above, I believe it is appropriate to consider Cisplatin + Etoposide or Cisplatin + Irinotecan as the evidence-based treatments, rather than Carboplatin+Etoposide/Irinotecan.
5. Second-line treatmen for EP-PD-NEC in the advanced setting
Page 4 Line 159-Page 5 Line 181; The authors' statement in this section pertains to Small Cell Lung Cancer (SCLC) and not specifically to EP-PD-NEC. While SCLC and EP-PD-NEC are generally considered to be biologically similar, it is important to include evidence specifically related to EP-PD-NEC and not rely on citations from studies targeting a different condition.
Author Response
Summary
Weaver et al. summarized a review on the chemotherapy for Extra-pulmonary Neuroendocrine Carcinoma (EP-PD-NEC). While this field has limited evidence, high-quality studies, including phase III trials, have been reported in recent years, highlighting the relatively high importance of this review. However, there are several concerns on this paper with the evaluation of the level of evidence, and major revisions are necessary.
Major points
- Introduction
Page 2 Line 45-48; The authors have described the significance of TP53 mutations and RB1 loss in poorly differentiated neuroendocrine tumors. While they have cited the paper by van Riet et al., there are other important studies worth considering as well.
One example is the following:
Hijioka et al. Pancreatology. 2020 Oct;20(7):1421-1427. doi: 10.1016/j.pan.2020.07.400. Epub 2020 Aug 8.
Authors: Thank you for pointing out this reference we have added this reference to the relevant section of the text.
- Introduction
Page 2 Line 62-63; In this section, the authors mention that a Ki-67 index of higher than 55% indicates a good response to first-line chemotherapy for EP-PD-NEC. However, it is important to note that the cited paper by Sorbye et al. is predating the introduction of the NET G3 concept and the current definition of EP-PD-NEC. Furthermore, it should be clarified that the favorable response applies specifically to cytotoxic chemotherapy and does not always imply a good response to all types of chemotherapy (Hijioka et al. Clin Cancer Res
. 2017 Aug 15;23(16):4625-4632. doi: 10.1158/1078-0432.CCR-16-3135. Epub 2017 Apr 28.)
Authors: Again thank you for pointing this out. We have amended the text to make it clear that we are referring to cytotoxic chemotherapy and have included a reference to this paper, discussing the fact that associations with Ki67 may be confounded in earlier studies.
- Introduction
Page 2 Line 71-75; Regarding the proposed algorithm for chemotherapy in EP-PD-NEC in Figure 1, I cannot agree with its content. Firstly, regarding first-line chemotherapy, there is limited high-quality evidence for Carboplatin + Etoposide/Irinotecan. Most studies are retrospective, and there is a lack of randomized prospective double-blinded trials. the Cisplatin + Etoposide/Irinotecan, which was demonstrated in the TOPIC-NEC study that there was equivalence efficacy. Carboplatin is considered an alternative treatment option in limited situations, such as when Cisplatin is not suitable due to impaired renal function, etc.
Authors: This is an important point. We have amended the text and figure to make it clear that in light of the TOPIC-NEC data if a patient is fit and young then cisplatin based treatment should be considered over carboplatin based. ENETs 2023 guidelines state that carboplatin or cisplatin are interchangeable based on retrospective data and implications form SCLC data but as the reviewer suggests the only randomised data available utilised cisplatin. In practice given the aggressive nature of this tumour and short OS many patients are not fit for cisplatin based treatment and consideration should be given to which regimen will maximise QoL and we hope the changes we have made to the review reflect this equipoise.
For second-line chemotherapy, the algorithm suggests TemCap is effective for Ki-67 indices ranging from 20-54%, however, it should actually be based on a cut-off of 55% and it should be 20-55%. Additionally, I believe that there is limited evidence supporting the efficacy of TemCap specifically for NEC, rather than NET-G3.
Authors: We have followed the range given in the recent ENETS 2023 guidelines for consideration of temcap theapy in patients with lower proliferation defined as <55%. We have modified our figure to make this clearer and to emphasise the reviewers point. Evidence is limited in NEC but but hope we have summarised it in the relevant section on temozolomide monotherapy and temcap dual therapy. We have added in an additional reference as well in a section on MGMt methyltion requested by another reviewer.
- First line systemic therapy options for patients with advanced EP-PD-NEC
Page 3 Line 83; As mentioned above, I believe it is appropriate to consider Cisplatin + Etoposide or Cisplatin + Irinotecan as the evidence-based treatments, rather than Carboplatin+Etoposide/Irinotecan.
Authors: As above we have now addressed this important point
- Second-line treatmen for EP-PD-NEC in the advanced setting
Page 4 Line 159-Page 5 Line 181; The authors' statement in this section pertains to Small Cell Lung Cancer (SCLC) and not specifically to EP-PD-NEC. While SCLC and EP-PD-NEC are generally considered to be biologically similar, it is important to include evidence specifically related to EP-PD-NEC and not rely on citations from studies targeting a different condition.
Authors: Thank you for this point. We agree there is clearly a difference between these entities at that clinically relevant findings in SCLC cannot be transferred directly to EP-PD-NEC. However we do believe the Hadoux et al data which includes EP-PD-NEC provides some evidence that earlier rechallenges could be considered. We have added text to this section and the figure to clarify the uncertainty here.
Round 2
Reviewer 2 Report
1.The single agent lurbinectedin section (line 340) is listed under the combination 2L chemotherapy header--suggest moving to single agent section
2. Table 2—would include P values as appropriate
3. Line 426—BRAF inhibitor section-- There is no mention of combination therapy with MEK inhibitors. Not only does the combination have tumor agnostic approval-- the paper that the authors reference (Klemper et al ) refers to use of BRAF-MEK combination therapy.
Author Response
1.The single agent lurbinectedin section (line 340) is listed under the combination 2L chemotherapy header--suggest moving to single agent section
Authors: Thank you for spotting this error, we have moved it to the appropriate section now.
- Table 2—would include P values as appropriate
Authors: Thank you for pointing this out. We have amended the table legend to make it clear that Bevanec and NET-02 were non-comparative trials so no p-values are available for the table and that TOPIC-NEC found no significant difference between any of the comparisons.
- Line 426—BRAF inhibitor section-- There is no mention of combination therapy with MEK inhibitors. Not only does the combination have tumor agnostic approval-- the paper that the authors reference (Klemper et al ) refers to use of BRAF-MEK combination therapy.
Authors: Thank you we have now included reference to the fact that the BRAF inhibitors are given with MEK inhibitors in the reference and in the tumour agnostic approval.
Reviewer 3 Report
This paper is a significant contribution, and I think the current revision can be accepted for publication.
Author Response
This paper is a significant contribution, and I think the current revision can be accepted for publication.
Authors: Thank you for your time reviewing our manuscript.